# Flexible and Reusable Ag Coated TiO_2_ Nanotube Arrays for Highly Sensitive SERS Detection of Formaldehyde

**DOI:** 10.3390/molecules25051199

**Published:** 2020-03-06

**Authors:** Tong Zhu, Hang Wang, Libin Zang, Sila Jin, Shuang Guo, Eungyeong Park, Zhu Mao, Young Mee Jung

**Affiliations:** 1School of Chemistry and Life Science, Advanced Institute of Materials Science, Changchun University of Technology, Changchun 130012, China; 2Department of Chemistry, Institute for Molecular Science and Fusion Technology, Kangwon National University, Chuncheon 24341, Korea; zanglb13@jlu.edu.cn (L.Z.); jsira@kangwon.ac.kr (S.J.); guoshuang0420@163.com (S.G.); egpark@kangwon.ac.kr (E.P.)

**Keywords:** formaldehyde, SERS, Raman, semiconductor, TiO_2_

## Abstract

Quantitative analysis of formaldehyde (HCHO, FA), especially at low levels, in various environmental media is of great importance for assessing related environmental and human health risks. A highly efficient and convenient FA detection method based on surface-enhanced Raman spectroscopy (SERS) technology has been developed. This SERS-based method employs a reusable and soft silver-coated TiO_2_ nanotube array (TNA) material, such as an SERS substrate, which can be used as both a sensing platform and a degradation platform. The Ag-coated TNA exhibits superior detection sensitivity with high reproducibility and stability compared with other SERS substrates. The detection of FA is achieved using the well-known redox reaction of FA with 4-amino-3-hydrazino-5-mercapto-1,2,4-triazole (AHMT) at room temperature. The limit of detection (LOD) for FA is 1.21 × 10^−7^ M. In addition, the stable catalytic performance of the array allows the degradation and cleaning of the AHMT-FA products adsorbed on the array surface under ultraviolet irradiation, making this material recyclable. This SERS platform displays a real-time monitoring platform that combines the detection and degradation of FA.

## 1. Introduction

Formaldehyde (FA), a colorless molecule that is a strong irritant, is a major hazard to human health and has been identified as a carcinogen [1,2,3]. The commercial product is an aqueous solution, and a 35–40% aqueous solution is commonly called formalin [4,5]. At present, many FA detection methods have been studied, including high-performance liquid chromatography (HPLC) [6], gas chromatography-mass spectrometry (GC-MS) [7], and fluorescence analysis [8]. However, these methods have certain disadvantages, such as a long detection time, complicated preprocessing steps, and incomplete detection ability. Therefore, it is important to establish a simple, fast, green, sensitive, and selective trace FA detection method.

Raman spectroscopy is an important analytical technique that provides molecular information by obtaining structural fingerprints of molecular vibrational levels [9,10]. However, Raman spectroscopy has the disadvantages of very weak signals and low reproducibility due to the inelastic scattering of radiation. Surface-enhanced Raman scattering (SERS), a fast, sensitive, and nondestructive spectroscopy tool for identifying and detecting chemical/biological species, has received considerable attention because it significantly amplifies the effect of adsorbed molecules on the target Raman signal [11,12,13,14,15,16,17,18]. SERS was first used in 1973 with pyridine adsorbed on rough silver [19]. Since its discovery, SERS has attracted widespread attention, and many studies on SERS have been launched and are ongoing. SERS has shown great advantages in terms of high selectivity, fluorescence quenching ability, and nonphotodegradation of molecules. In recent years, semiconductor nanoarrays modified with metal nanoparticles have been used to generate high-density internal hot spots to improve the uniformity and detection sensitivity of the substrate [20,21,22,23]. In this work, we have developed an SERS method for the detection of target molecules (such as FA) on metal-semiconductor substrates. The use of SERS technology is expected to solve the above problems of FA detection.

Recently, photocatalytic organic molecular processes have also been successfully detected by SERS. Semiconductor-based composite materials including Ag/TiO_2_, Au/TiO_2_, Ag/SiO_2_, Ag/RGO, Au/ZnO, Cu_2_O/Ag, etc., have been shown to have both SERS activity and photocatalytic degradation activity [24,25,26,27,28,29,30,31,32,33,34,35,36,37,38,39,40,41]. In particular, semiconductor structures based on one-dimensional arrays have been proven to serve as excellent SERS substrates with higher sensitivity, uniformity, and reproducibility than other structures. However, semiconductor surface cleanliness is particularly important for overall SERS performance. Due to the limitations of the synthesis methods, most metal-semiconductor composite substrates cannot avoid factors that degrade SERS performance, such as surface functionalization and the occurrence of residues of synthetic materials during the synthesis procedure. In addition, Ag or Au nanoparticles in most metal-semiconductor composites show extremely uncontrollable aggregation, which greatly reduces the repeatability of SERS detection. Therefore, it is still challenging to prepare a metal-semiconductor SERS substrate with a clean surface and high repeatability.

TiO_2_ nanotube array structures have been shown to have high light scattering efficiency, high surface cleanliness, high specific surface area, and excellent electron transport properties. These structures also provide greater flexibility and maneuverability than other structures for use as SERS substrates. In this work, the TiO_2_ nanotube array, prepared based on the anodic oxidation method, has a relatively clean surface, which is conducive to absorbing a large amount of the target on the surface. Moreover, we combine the highly uniformly distributed liquid, Ag sol, with the highly ordered TiO_2_ nanotube array (TNA) structure, thereby greatly improving the SERS repeatability of the overall substrate, compared with the repeatability of other substrates. At the same time, the substrate retains the recyclability and SERS activity for photocatalytic degradation inherent in the Ag-TiO_2_ structure.

In this study, we first prepared an ideal TNA by optimizing the experimental conditions. Silver sol was then used to adsorb silver nanoparticles on the TNA to increase the SERS signal. Finally, 4-amino-3-hydrazino-5-mercapto-1,2,4-triazole (AHMT) was used as a Raman probe to sensitively detect FA, and to simultaneously reveal the reusability of the SERS substrate and the ability to degrade organic FA [42]. In summary, we have prepared a new type of SERS substrate with high sensitivity and selective recognition of FA; the substrate is inexpensive, environmentally friendly and reusable.

## 2. Results and Discussion

Figure 1A shows scanning electron microscopy (SEM) images of the TNA material prepared by surface pretreatment of pure titanium foil through two anodizing methods in a 0.5 wt% NH_4_F ethylene glycol system at a controlled voltage of 60 V. From the top view, it can be seen that the surface of the TNA consists of a nanotube array, and the diameter of the tubes is 100–200 nm. From the side view, it can be seen that the TNA is open at the top and closed at the bottom. At the bottom of the film, there is a thin and dense barrier layer. The barrier layer is uniformly distributed, perpendicular to the substrate nanotube array, and the tube length is 4–7 μm.

Figure 1B shows the XRD patterns of TiO_2_ nanotubes produced at 60 V with 120 min of oxidation time. The eight sharp peaks at 2*θ* = 25.2°, 47.9°, 53.8°, 55.0°, 62.8°, 68.8°, 70.4°, and 74.9° were indexed to anatase TiO_2_ (101), (200), (105), (211), (204), (116), (220), and (215), respectively. The results indicated that the prepared TiO_2_ nanotubes are in anatase crystal form. The distribution of Ag over the nanotubes is shown in Appendix A. Most Ag nanoparticles were distributed on the TNA surface. During the preparation of SEM samples, Ag may aggregate during the drying step. The SERS test can be completed in 1 min; during this time, the Ag nanoparticles are moving, and their distribution is more uniform than that shown in the figure.

Figure 2 shows the preliminary preparation scheme in our experiment. The figure illustrates the preparation of the TNA on titanium foil by the anodization method. Silver nanoparticles were applied to the TNA by coating a silver sol to ensure the formation of an SERS substrate with the attached precious metal. Studies have reported that Ag-TiO_2_ composites can be used to detect and degrade organic pollutants. We optimized an Ag-TiO_2_ material to explore whether the detection of FA was satisfactory.

In this work, the SERS method is based on the reaction of excess AHMT with FA (Figure 3A). Under basic conditions, FA and AHMT undergo a condensation reaction to form 6-mercapto-5-triazolo [4,3-b]-s-tetrazine (MTT). It can be seen from Figure 3B that with a decrease in FA concentration, the characteristic band intensity in the ultraviolet-visible (UV-vis) absorption spectrum has a tendency to first increase and then decrease. An FA concentration of 10^−4^ M cannot be detected. Therefore, the sensitivity of UV-vis spectroscopy to FA is not ideal. Thus, we prepared a recyclable SERS substrate with photocatalytic performance to detect FA. We also investigated the reproducibility, stability, and recycling characteristics of the SERS substrate for use as a catalyst. To examine these properties, we performed the following experiments.

Figure 4A shows the SERS spectra of a mixture of AHMT and AHMT-MTT in the 600–1800 cm*^−^*^1^ region using the Ag-sol substrate. Figure 4B shows the scheme of the SERS sample deposition and measurement. As shown in Table 1, the bands at 710 and 832 cm*^−^*^1^ are due to the S-C-N tensile vibration and N-C-N tensile vibration of the AHMT and MTT molecules, respectively. The six rings of MTT make its N-C-N tensile vibration stronger than that of AHMT. The SERS band at 1473 cm*^−^*^1^ is significantly enhanced, which is attributed to the tensile vibration mode of C-C rings. The other main bands of AHMT are observed at 1217, 1286, and 1391 cm*^−^*^1^, which are attributed to in-ring breathing vibrations and in-plane deformation. Compared to AHMT, several bands of MTT are blueshifted due to the six rings.

Figure 4C shows the SERS spectra of AHMT-FA with various concentrations of FA (from 1.44 × 10^−2^ to 1.44 × 10^−9^ M). With decreasing FA concentration, the intensity of the Raman band at 1473 cm^−1^ greatly decreased, and the band was clearly observed down to an FA solution concentration of 1.44 × 10^−9^ M, relative to the band of the probe solution (AHMT). Therefore, this method is reasonable and suitable for the detection of FA with a detection limit of 1.44 × 10^−9^ M. Figure 5 shows the Raman intensity of the characteristic band at 1286 cm^−1^ at different concentrations. The test was repeated six times for each concentration. The results also show the average and standard deviation for each concentration during the test. The accuracy of the SERS method for detecting FA is shown. It can be seen from Figure 5 that as the MTT concentration decreases, the average value and standard deviation also decrease. The results show that the SERS method is sensitive to the detection of FA and further prove the feasibility of this method.

The calculated limit of detection (LOD) is given by the black line. The standard curve of the AHMA-FA solution obeys the equation Y = 5.1399 + 0.2701 X, which shows a linear relationship and has a correlation coefficient of 0.9639. The limit of detection (LOD) was determined according to the IUPAC recommendations (the minimally acceptable signal intensity must be three times greater than the standard deviation of the blank signal). As depicted in Figure 5, this intensity approximately corresponds to 10^−6.92^ M (i.e., 1.21 × 10^−7^ M). Finally, in terms of reproducibility and repeatability, SERS measurements were obtained on 20 spots, randomly selected on the Ag-TNA substrates. (see Appendix A).

The degradation process during the experiments was monitored by the changes in the band intensity of AHMT-FA in the SERS spectra. Taking 1.44 × 10^−4^ M as an example, as shown in Figure 6A, lines a–h correspond to Ag-TNA substrates used to adsorb AHMT-FA and irradiated under ultraviolet light at 254 nm for 40, 50, 60, 70, 100, 130, 160, and 190 min, respectively. Line i is the Raman spectrum of Ag-TNA without adsorbed probe molecules. The comparison shows that the probe molecules, AHMT and FA, were completely degraded after approximately 3 h of irradiation. This result shows that the Ag-TNA substrate has a self-cleaning function under ultraviolet irradiation, and this function can overcome the disadvantage of being able to use SERS substrates only once.

As mentioned in the introduction, organic molecules on the surface of TiO_2_ can be degraded by the generation of free radicals and oxidizing substances on the surface of TiO_2_ under ultraviolet irradiation. In visible light, the localized surface plasmon resonance (LSPR) of AgNPs is excited and decays to generate hot electrons. The energy of these electrons is higher than the potential barrier between TiO_2_ and AgNPs, and the electrons can jump to the conduction band of TiO_2_ and generate holes in AgNPs. The interface state density of the TNA and Ag is relatively large, and most of the hot electrons are captured by the interface state, thus greatly reducing the total number of electrons that can reach the conduction band of TiO_2_. The remaining electrons that reach the conduction band of TiO_2_ migrate to the surface of TiO_2_, combine with molecular oxygen adsorbed on the surface to form · O^2−^, or combine with ·OH, resulting from the decomposition of H_2_O in the oxygen vacancy to form ·OH. Note that ·O^2−^ and ·OH have strong catalytic activity and are the main active species for catalyzing the degradation of organic matter. Due to the characteristics of TiO_2_ nanotubes, they have a higher electron transport efficiency than other compounds, which is conducive to the transfer of hot electrons from AgNPs to the conduction band of TiO_2_, thus increasing the photocatalytic efficiency.

Figure 6C shows the intensity of the characteristic band at 1286 cm^−1^ as a function of degradation time. The results show that as the degradation time increases, the intensity of the Raman band significantly decreases, and the overall Raman signal intensity decreases. Furthermore, after detecting FA, the reversible SERS active substrate could degrade the FA and AHMT adsorbed on the substrate surface into small inorganic molecules and water molecules by photocatalytic degradation (Figure 6B). This result further proves the self-cleaning function.

To evaluate the degradation performance and recyclability of this Ag-TNA substrate, we performed three detection-degradation cycle experiments on the Ag-TNA substrate. We used 1.44 × 10^−4^ M FA as an example (Figure 7). In the first cycle, line a represents the Raman spectrum of AHMT-FA reacting for 16 h (first reaction), and line b represents the Raman spectrum after 3 h of 254 nm irradiation (first ultraviolet irradiation). In the second cycle, line c indicates the Raman spectrum of AHMT-FA reacting for 16 h using the substrate in line b (second reaction), and line d indicates the Raman spectrum after 3 h of 254 nm irradiation (second ultraviolet irradiation). In the third cycle, line e indicates the Raman spectrum of AHMT-FA reacting for 16 h (third reaction), and line f indicates the Raman spectrum after 3 h of 254 nm irradiation (third ultraviolet irradiation). The Ag-TNA SERS substrate used in this experiment can be completely reused. This figure also shows how a reversible SERS-active substrate works; after detecting FA via SERS, the substrate can photocatalytically degrade FA and AHMT adsorbed on the substrate surface, exhibiting a self-cleaning function. This result provides prospects for the on-site detection and degradation of organic pollutants by SERS. At the same time, FA is a major indoor pollutant. The existing commercial FA detection method is mainly based on electrochemical sensing technology, and its shortcomings are low specificity and an insufficient ability to discriminate volatile organic compounds (VOCs). In this work, the preparation method of Ag-TNA is simple, and it is easy to prepare in a large area. At the same time, Ag-TNA is an excellent SERS substrate for FA detection with good SERS activity and photocatalytic degradation performance. In the future, we will try to detect gaseous formaldehyde based on this work. This method has great potential to complement the selectivity of existing formaldehyde gas detection methods.

## 3. Materials and Methods

### 3.1. Chemicals

AHMT was purchased from Sigma-Aldrich Co., Ltd. (St. Louis, MO, USA) and was used without further purification. Titanium foil and silver nitrate (AgNO_3_) were also purchased from Sigma-Aldrich Co., Ltd. (St. Louis, MO, USA). FA (40%) was purchased from Liaoning Quanrui Reagent Co., Ltd. Hydrofluoric acid, ammonium fluoride, and ethylene glycol (analytical reagents) were purchased from Aladdin Company and were used without further purification. The water used in the experiment was ultrapure water, and the ethanol used was anhydrous ethanol.

### 3.2. Instruments

SEM characterization was performed by a JEOL 7610 p thermal field emission scanning electron microscope. UV-vis absorption spectra were recorded on a Cary 5000 ultraviolet–visible–near infrared (UV–VIS–NIR) spectrometer (Agilent Technologies, Inc., Santa Clara, CA, USA). XRD testing was conducted by a Rigaku Smartlab X-ray diffractometer using CuKα radiation (λ = 1.5418 Å) at 45 kV and a 200 mA excitation light source. Raman spectroscopy was performed using a Jobin Yvon/Horiba LabRAM HR Evolution confocal micro-Raman spectrometer equipped with a multichannel air-cooled charge-coupled device (CCD) detector. A 532 nm laser was used as the excitation light source. A monocrystalline silicon wafer was used to calibrate the Raman spectrometer. All Raman spectra were measured with a 600 g/mm grating using a 50×, 0.50 NA (OLYMPUS LMPlanFLN, Tokyo, Japan) long-working-distance (LWD) microscope objective. The laser power reaching the sample surface was approximately 5 mW. The acquisition time of Raman spectra was 30 s for each window. For ultraviolet light degradation, a ZF-7 portable ultraviolet analyzer (including 254 nm and 365 nm ultraviolet radiation) produced by Shanghai Daluo Scientific Instrument Co., Ltd. (Shanghai, China) was used. The UV degradation experiment was performed under irradiation with a 254 nm UV lamp with a power of 8 W.

### 3.3. Preparation of Silver Sol

AgNO_3_ (169 mg) was added to a three-necked flask, 100 mL of deionized water and a stir bar were added, and a thermometer and a condenser tube were connected. Sodium citrate (0.01 g) was added to 1 mL of deionized water. Good silver sol was obtained by heating the AgNO_3_ and deionized water until slightly boiling, dropwise adding sodium citrate (discoloration within 3 min), and heating for 30 min.

### 3.4. Preparation of the TNA

Titanium foil with a size of 4 × 5 cm was used as the raw material. The surface of the titanium foil was cleaned with acetone, isopropanol, methanol, and ultrasonication. Then, the surface was washed with deionized water and blown with nitrogen before use. Graphite flakes (also ultrasonically cleaned with distilled water) were used as the cathode, and titanium foil was used as the anode. Anode oxidation was performed at room temperature, and 0.5 wt% NH_4_F glycol was used as the electrolyte. After the electrolytic titanium dioxide was washed with distilled water, it was soaked in water; the titanium dioxide film formed on the surface, was peeled off in the water and dried under strong nitrogen flow, and a second round of electrolysis was performed for 2 h. The electrolyzed titanium foil was washed with absolute ethanol and ultrapure water, soaked in absolute ethanol overnight, removed under strong nitrogen flow, blown dry, and calcined in a muffle furnace at 450 °C for 2 h. The TNA was stored in a box protected from light.

### 3.5. Preparation of the Test Solution

At room temperature, 0.0124 g AHMT was added into a 50 mL centrifuge tube. Then, 25 mL of deionized water and 25 mL of 5.0 M NaOH were added to yield a probe solution with a concentration of 3.34 × 10^−3^ M. The concentration of the FA (40%) solution was 14.4 M, which was sequentially diluted to 10^−9^ M. One milliliter of each FA solution and 2 mL of probe solution were mixed and reacted at room temperature for 30 min to obtain the test liquids (FA concentrations of 1.44, 1.44 × 10^−1^, 1.44 × 10^−2^, 1.44 × 10^−3^, 1.44 × 10^−4^, 1.44 × 10^−5^, 1.44 × 10^−6^, 1.44 × 10^−7^, 1.44 × 10^−8^, and 1.44 × 10^−9^ M).

### 3.6. SERS Detection of FA

The calcined TNA was placed into the test liquid for 16 h. Then, the TNA was removed, 150 μL silver sol was added, and the confocal micro-Raman spectrometer with a 532 nm excitation laser was used to examine the test liquids with different concentrations of FA. The Raman signal of the probe (AHMT concentration of 3.34 × 10^−3^ M) was detected by the same method.

### 3.7. UV Degradability of AHMT-FA with the Ag-TNA

A test liquid with a concentration of 1.44 × 10^−4^ M was selected to evaluate the ultraviolet photodegradability of the SERS substrate. First, the TNA was immersed in the 1.44 × 10^−4^ M test solution for 16 h, and then the TNA was removed. Silver sol was added to measure its initial Raman signal; then, it was directly put into a portable ultraviolet analyzer and irradiated with 254 nm ultraviolet light for 3 h. The Raman signal was detected every 10 or 30 min until the degradation was complete.

### 3.8. Recyclable SERS Substrate

(1) As the degradation time increased, the Raman intensity continued to decrease until the Raman signal basically disappeared; this time corresponded to the time required to obtain complete degradation of AHMT-FA. (2) The same TNA was put into the test solution again for 16 h. Then, the silver sol was removed, and the Raman signal of AHMT-FA was detected. (3) The solution was placed under ultraviolet light with a wavelength of 254 nm and irradiated to complete degradation. (4) The experiment was repeated one more time at the end for a total of three replicates to check the substrate recyclability.

## 4. Conclusions

In this study, we show a new method for FA detection based on SERS. The combination of a surface-cleaned TNA array material prepared by electrochemical anodization and Ag-sol constitutes a dual-function Ag-TNA FA-sensing material that has a SERS effect and photocatalytic degradation. The results show that the SERS intensity of the FA reaction product has a strong FA concentration dependence, and that the lowest detection concentration of FA in solution can reach 10^−9^ M. Furthermore, it has been proven that the composite array is a highly sensitive, stable, self-cleaning and recyclable SERS-active substrate. In addition, the excellent photocatalytic degradation performance and SERS activity of the substrate show great potential for the on-site detection and degradation of organic pollutants by the SERS method.

## Figures and Tables

**Figure 1 molecules-25-01199-f001:**
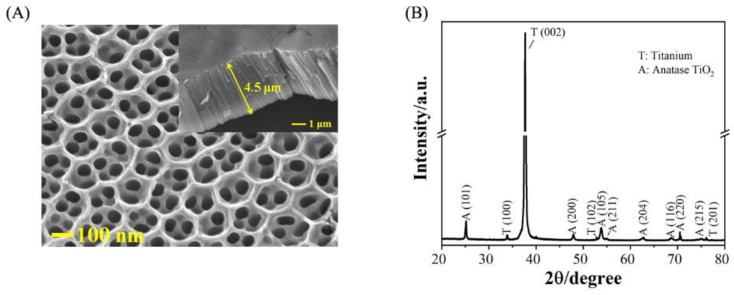
(**A**) Top-view and side-view SEM images of the TiO_2_ nanotube array (TNA) and; (**B**) XRD pattern of the TNA calcined at 450 °C.

**Figure 2 molecules-25-01199-f002:**
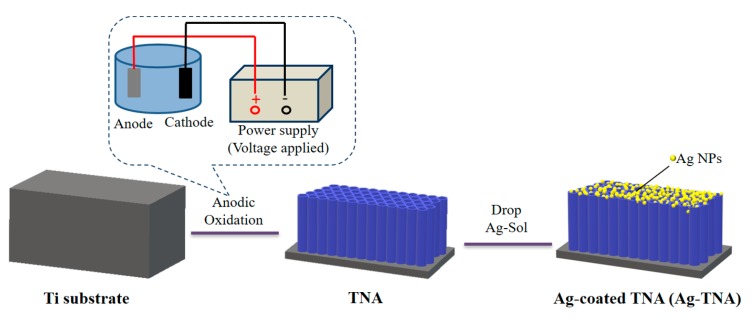
Illustration of TNA and Ag-TNA preparation.

**Figure 3 molecules-25-01199-f003:**
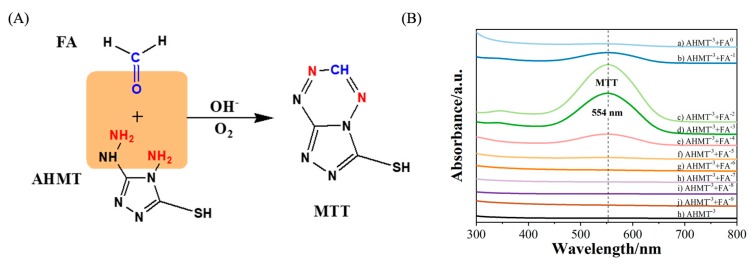
(**A**) Derivative reaction of formaldehyde (FA) and 4-amino-3-hydrazino-5-mercapto-1,2,4-triazole (AHMT) under alkaline conditions. (**B**) UV-vis absorption spectra of different concentrations of FA after reaction with AHMT. (**a**–**j**) FA (1.44, 1.44 × 10^−1^, 1.44 × 10^−2^, 1.44 × 10^−3^, 1.44 × 10^−4^, 1.44 × 10^−5^, 1.44 × 10^−6^, 1.44 × 10^−7^, 1.44 × 10^−8^, and 1.44 × 10^−9^ M) + AHMT (3.34 × 10^−3^ M); (**h**) AHMT (3.34 × 10^−3^ M).

**Figure 4 molecules-25-01199-f004:**
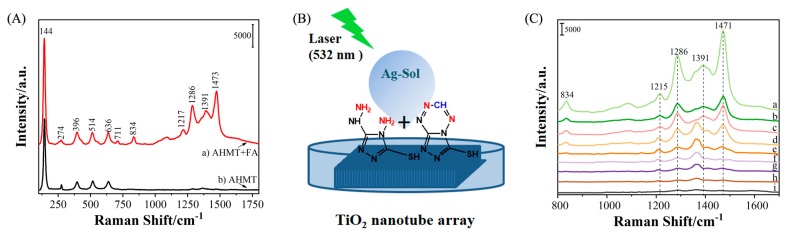
(**A**) Surface-enhanced Raman scattering (SERS) spectra of AHMT and AHMT-FA (6-mercapto-5-triazolo [4,3-b]-s-tetrazine,MTT): (**a**) A mixture of AHMT and MTT (with an excess of AHMT in the reaction.); (**b**) AHMT (3.34 × 10^−3^ M). (**B**) SERS detection process diagram. (**C**) SERS detection of FA (from 1.44 × 10^−2^ to 1.44 × 10^−9^ M) using AHMT as a probe (3.34 × 10^−3^ M) after 16 h of reaction. (**a**–**h**) FA (1.44 × 10^−2^–1.44 × 10^−9^ M) + AHMT (3.34 × 10^−3^ M) and (**i**) AHMT (3.34 × 10^−3^ M).

**Figure 5 molecules-25-01199-f005:**
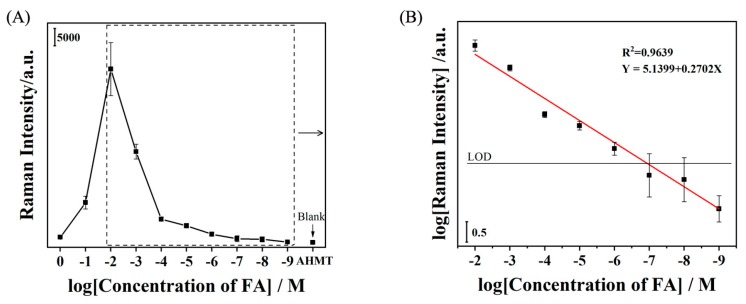
Concentration-dependent curve for the SERS detection of FA using AHMT as a probe (3.34 × 10^−3^ M). (**A**) Intensity of the Raman band at 1286 cm^−1^ plotted vs the concentration of FA (from 1.44 to 1.44 × 10^−9^ M). (**B**) Linear relationship between the logarithmic Raman intensity and logarithmic concentration of FA (from 1.44× 10^−2^ to 1.44 × 10^−9^ M).

**Figure 6 molecules-25-01199-f006:**
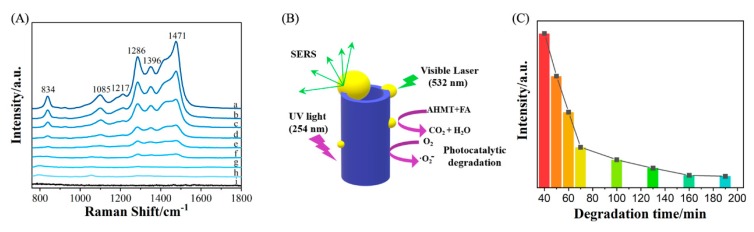
(**A**) SERS spectra of AHMT + MTT on the Ag-coated TNA substrate surface after (**a**) 40, (**b**) 50, (**c**) 60, (**d**) 70, (**e**) 100, (**f**) 130, (**g**) 160 and (**h**) 190 min of catalytic degradation by UV irradiation; (**i**) Raman spectrum of the Ag-TNA substrate without adsorbate. (**B**) Schematic of UV-degradable organics on the SERS substrate. (**C**) Plot of the Raman band intensity of 1.44 × 10^−4^ M MTT at 1286 cm^−1^ as a function of degradation time.

**Figure 7 molecules-25-01199-f007:**
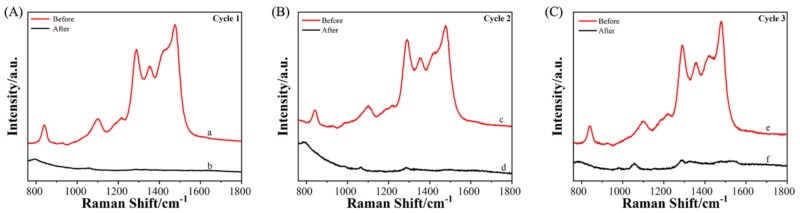
SERS substrate for recyclable FA detection (three cycles of FA detection). (**A**) First cycle; (**B**) second cycle; and (**C**) third cycle. The red and black spectra represent the detection of FA and the degradation after UV light irradiation, respectively.

**Table 1 molecules-25-01199-t001:** Vibration modes represented by AHMT-FA characteristic bands.

Functional Group/Vibration	Raman Bands of AHMT-FA
E_g_	144, 636 cm^−1^
B_1g_	396 cm^−1^
A_1g_/B_1g_	514 cm^−1^
ν(S-C-N)	710 cm^−1^
ν(N-C-N)	832 cm^−1^
ν(C-C)	1473 cm^−1^
Breathing vibration	1217, 1286, 1391 cm^−1^

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
