# Peer review of "Flexible and Reusable Ag Coated TiO2 Nanotube Arrays for Highly Sensitive SERS Detection of Formaldehyde"

_molecules, 2020, doi:10.3390/molecules25051199_

Round 1
Reviewer 1 Report
The manuscript "Flexible and reusable Ag coated TiO2 nanotube arrays for highly sensitive SERS detection of formaldehyde", written by Zhu et al., describes a new method for the analysis of Formaldehyde using surface enhahced Raman spectroscopy. The theme is interesting and the developed method has a good application potential.
I do have following comments:
- The distribution of Ag over the nanotubes filed should be better characterized using for example EDX mapping or similar technique.
- Data describing method repeatability and reproducibility should be added.
- Error bards should be added into he figure 5B
Author Response
We appreciate the Reviewer’s comments on this manuscript, which helped us revise the manuscript significantly. All the replies are written in boldface.
Response to Reviewer #1
The manuscript "Flexible and reusable Ag coated TiO2 nanotube arrays for highly sensitive SERS detection of formaldehyde", written by Zhu et al., describes a new method for the analysis of Formaldehyde using surface enhanced Raman spectroscopy. The theme is interesting and the developed method has a good application potential.
I do have following comments:
The distribution of Ag over the nanotubes filed should be better characterized using for example EDX mapping or similar technique. The distribution of Ag over the nanotubes filed should be better characterized using for example EDX mapping or similar technique.
According to the Reviewer’s comment, we characterized the distribution of Ag on the nanotubes using SEM and EDX mapping techniques. We have added the data as “Figure S1” in the “Supplementary Material file” and provided the corresponding explanations in the revised manuscript.
The distribution of Ag on the nanotubes is shown in Figure S1. Most Ag nanoparticles were distributed on the TNA surface. During the preparation of SEM samples, Ag may aggregate during the drying process. The SERS test can be completed in 1 min; during this time, the Ag nanoparticles are moving, and their distribution is more uniform than that shown in Figure S1.
These revised sentences appear in the Results and Discussion section on lines 97-101 on page 3 of the revised manuscript (when the "Track Changes" is displayed).
Data describing method repeatability and reproducibility should be added.
According to the Reviewer’s comment, we have added the data that show the repeatability and reproducibility of the method Figure S2 of the Supplementary Material.
The standard curve of the AHMA-FA solution obeys the equation Y = 5.1399 + 0.2701 X, which shows a linear relationship and has a correlation coefficient of 0.9639. The limit of detection (LOD) was determined according to the IUPAC recommendations (the minimally acceptable signal intensity must be three times greater than the standard deviation of the blank signal). As depicted in Figure 5, this intensity approximately corresponds to 10−6.92 M (i.e., 1.21´10−7 M). Finally, in terms of reproducibility and repeatability, SERS measurements were obtained on 20 spots randomly selected on the Ag-TNA substrates (see Figure S2 in the Supplementary Material).
These revised sentences appear in the Results and Discussion section on lines 155-162 on page 5 of the revised manuscript (when the "Track Changes" is displayed).
Error bards should be added into the figure 5B.
According to the Reviewer’s comment, we have added error bars to Figure 5B in the revised manuscript.

Reviewer 2 Report
The paper is written in a good English, well organized with a logical progression of thought within each section. Authors clearly identify the key information obtained and illustrate in a comprehensive manner the advancement of knowledge and new insights gained with the research.
I suggest to more articulate discussion of the implications about the findings presented, including future directions of research and/or translational potential.
Figures/tables well-structured and self-explanatory. However, I suggest an extensive and more articulate critical analysis of the results found by the authors in the light of the research questions and of the existing literature. To this scope, it is necessary to report, in the "Introduction section" , many other specific informations about the detection of FA, mainly by SERS approach but using engigneering different nanosystems (TiO2 nanoparticles, Ag/SiO2, graphene oxide (rGO)/[Ag(NH3)2]+ (rGO/[Ag(NH3)2]+/Atp) ETC); so, a comparison in terms of systems reproducibility, recycling, SERS enhancements etc will be possible in order to evidence the importance of author's results.
Author Response
We appreciate the Reviewer’s comments on this manuscript, which helped us revise the manuscript significantly. All the replies are written in boldface.
Response to Reviewer #2
The paper is written in a good English, well organized with a logical progression of thought within each section. Authors clearly identify the key information obtained and illustrate in a comprehensive manner the advancement of knowledge and new insights gained with the research.
I suggest to more articulate discussion of the implications about the findings presented, including future directions of research and/or translational potential.
According to the Reviewer’s comment, we supplied a more articulate discussion of the implications about the findings presented, including the mechanism of recyclable SERS-based FA detection, future directions of research and translational potential.
- Mechanism of recyclable SERS-based FA detection
As mentioned in the introduction, organic molecules on the surface of TiO2 can be degraded by free radicals and oxidizing substances generated on the surface of TiO2 under ultraviolet irradiation. In visible light, the localized surface plasmon resonance (LSPR) of AgNPs is excited and decays to generate hot electrons. The energy of these electrons is higher than the potential barrier between TiO2 and Ag NPs, and the electrons can jump to the conduction band of TiO2 and generate holes in Ag NPs. The interface state density of the TNA and Ag is relatively large, and most of the hot electrons are captured by the interface state, thus greatly reducing the total number of electrons that can reach the conduction band of TiO2. The remaining electrons that reach the conduction band of TiO2 migrate to the surface of TiO2, combine with molecular oxygen adsorbed on the surface to form · O2−, or combine with ·OH resulting from the decomposition of H2O in the oxygen vacancy to form ·OH. Note that ·O2− and ·OH have strong catalytic activity and are the main active species for catalyzing the degradation of organic matter. Due to the structural characteristics of TiO2 nanotubes, they have a higher electron transport efficiency than other compounds, which is conducive to the transfer of hot electrons from AgNPs to the conduction band of TiO2, thus increasing the photocatalytic efficiency.
- Future directions of research and translational potential
This result provides prospects for the on-site detection and degradation of organic pollutants by SERS. At the same time, FA is a major indoor pollutant. The existing commercial FA detection method is mainly based on electrochemical sensing technology, and its shortcomings are low specificity and insufficient ability to discriminate volatile organic compounds (VOCs). In this work, the preparation method of Ag-TNA is simple, and it is easy to prepare in a large area. At the same time, Ag-TNA is an excellent SERS substrate for FA detection with good SERS activity and photocatalytic degradation performance. In the future, we will try to detect gaseous formaldehyde based on this work. This method has great potential to complement the selectivity of existing formaldehyde gas detection methods.
These revised sentences appear in the Results and Discussion section on lines 178-191 on page 6 and lines 216-224 on pages 6-7 of the revised manuscript (when the "Track Changes" is displayed).
Figures/tables well-structured and self-explanatory. However, I suggest an extensive and more articulate critical analysis of the results found by the authors in the light of the research questions and of the existing literature. To this scope, it is necessary to report, in the "Introduction section" , many other specific informations about the detection of FA, mainly by SERS approach but using engigneering different nanosystems (TiO2 nanoparticles, Ag/SiO2, graphene oxide (rGO)/[Ag(NH3)2]+ (rGO/[Ag(NH3)2]+/Atp) ETC); so, a comparison in terms of systems reproducibility, recycling, SERS enhancements etc will be possible in order to evidence the importance of author's results.
According to the Reviewer’s comment, we conducted an extensive and critical analysis of our results based on the research questions and existing literature, as shown below.
(1) In the "Introduction" section, we have introduced in the literature the main FA detection methods in the literature based on SERS technology but using different nanosystems, such as those based on Ag/TiO2, Au/TiO2, Ag/SiO2, Ag/RGO, Au/ZnO, and Cu2O/Ag, and the FA detection method established by the SERS substrate, as discussed below.
Recently, photocatalytic organic molecular processes have also been successfully detected by SERS. Semiconductor-based composite materials including Ag/TiO2, Au/TiO2, Ag/SiO2, Ag/RGO, Au/ZnO, Cu2O/Ag, etc. have been shown to have both SERS activity and photocatalytic degradation activity [24-42]. In particular, semiconductor structures based on one-dimensional arrays have been proven to serve as excellent SERS substrates with higher sensitivity, uniformity, and reproducibility than other structures. However, semiconductor surface cleanliness is particularly important for overall SERS performance. Due to the limitations of the synthesis methods, most metal-semiconductor composite substrates cannot avoid factors that degrade SERS performance, such as surface functionalization and the occurrence of residues of synthetic materials during the synthesis procedure. In addition, Ag or Au nanoparticles in most metal-semiconductor composites show extremely uncontrollable aggregation, which greatly reduces the repeatability of SERS detection. Therefore, it is still challenging to prepare a metal-semiconductor SERS substrate with a clean surface and high repeatability.
(2) We further discussed these studies in comparison with our work in terms of system repeatability, recyclability, and SERS enhancement, and we analyzed the relative characteristics and importance of our results.
TiO2 nanotube array structures have been shown to have high light scattering efficiency, high surface cleanliness, high specific surface area, and excellent electron transport properties. These structures also provide greater flexibility and maneuverability than other structures for use as SERS substrates. In this work, the TiO2 nanotube array prepared based on the anodic oxidation method has a relatively clean surface, which is conducive to absorbing a large amount of the target on the surface. Moreover, we combine the highly uniformly distributed liquid Ag sol with the highly ordered TiO2 nanotube array (TNA) structure, thereby greatly improving the SERS repeatability of the overall substrate compared with the repeatability of other substrates. At the same time, the substrate retains the recyclability and SERS activity for photocatalytic degradation inherent in the Ag-TiO2 structure.
These revised sentences appear in the Introduction section on lines 54-75 on page 2 and references [24-42] (lines 380-425 on page 10-11) of the revised manuscript (when the "Track Changes" is displayed

Reviewer 3 Report
Authors report the preparation of a new SERS sensor for the detection of formaldehyde based on the TiO2-AgNPs materials. The utilized chemistry is simple and elegant and obtained results deserves attention and publication, however, before the publication a list of serious issues, connected with both materials characterization, as well as sensing properties should be solved:
- In the introduction section, authors write about SERS in general, however, there are a lot of reports about similar materials, J. Phys. Chem. C 2011, 115, 19, 9498-9502, Adv. Fun. Mat., 2010,20, 17,2815-2824. Please, discuss it and underline existing limitations
- About the characterization of material: it is absolutely not clear, whether AgNPs are just on the surface or they penetrate into TiO2 structure, what is the distribution (it affect the SERS response), loading and so on.. At least, show SEM and XRD data after AgNPs loading
- Experimental part: please, add parameters for Raman spectra collection (accumulation time and so on), what is power of UV lamp, which you used for degradation (this is an important parameter)
- Sensor: please, provide LOD determination according to IUPAC recommendations
Author Response
We appreciate the Reviewer’s comments on this manuscript, which helped us revise the manuscript significantly. All the replies are written in boldface.
Response to Reviewer #3
Authors report the preparation of a new SERS sensor for the detection of formaldehyde based on the TiO2-AgNPs materials. The utilized chemistry is simple and elegant and obtained results deserves attention and publication, however, before the publication a list of serious issues, connected with both materials characterization, as well as sensing properties should be solved:
In the introduction section, authors write about SERS in general, however, there are a lot of reports about similar materials, J. Phys. Chem. C 2011, 115, 19, 9498-9502, Adv. Fun. Mat., 2010,20, 17,2815-2824. Please, discuss it and underline existing limitations.
According to the Reviewer’s comment, we added two references (Ref [26]: J. Phys. Chem. C 2011, 115, 19, 9498-9502; Ref [32]: Adv. Fun. Mat., 2010,20, 17,2815-2824), which the reviewer provided and added other relevant references (Ref [24-42]) in the revised manuscript.
As the Reviewer pointed out, recently, photocatalytic organic molecular processes have also been successfully detected by SERS. Semiconductor-based composite materials including Ag/TiO2, Au/TiO2, Ag/SiO2, Ag/RGO, Au/ZnO, Cu2O/Ag, etc. have been shown to have both SERS activity and photocatalytic degradation activity [24-42]. In particular, semiconductor structures based on one-dimensional arrays have been proven to serve as excellent SERS substrates with higher sensitivity, uniformity, and reproducibility than other structures. However, semiconductor surface cleanliness is particularly important for overall SERS performance. Due to the limitations of the synthesis methods, most metal-semiconductor composite substrates cannot avoid factors that degrade SERS performance, such as surface functionalization and the occurrence of residues of synthetic materials during the synthesis procedure. In addition, Ag or Au nanoparticles in most metal-semiconductor composites show extremely uncontrollable aggregation, which greatly reduces the repeatability of SERS detection. Therefore, it is still challenging to prepare a metal-semiconductor SERS substrate with a clean surface and high repeatability.
TiO2 nanotube array structures have been shown to have high light scattering efficiency, high surface cleanliness, high specific surface area, and excellent electron transport properties. These structures also provide greater flexibility and maneuverability than other structures for use as SERS substrates. In this work, the TiO2 nanotube array prepared based on the anodic oxidation method has a relatively clean surface, which is conducive to absorbing a large amount of the target on the surface. Moreover, we combine the highly uniformly distributed liquid Ag sol with the highly ordered TiO2 nanotube array (TNA) structure, thereby greatly improving the SERS repeatability of the overall substrate compared with the repeatability of other substrates. At the same time, the substrate retains the recyclability and SERS activity for photocatalytic degradation inherent in the Ag-TiO2 structure.
We also supplemented the above discussion and highlighted the limitations of the existing reports and the importance of our work.
These revised sentences appear in the Introduction section on lines 54-75 on page 2 and references [24-42] (lines 380-425 on page 10-11) of the revised manuscript (when the "Track Changes" is displayed).
About the characterization of material: it is absolutely not clear, whether AgNPs are just on the surface or they penetrate into TiO2 structure, what is the distribution (it affect the SERS response), loading and so on. At least, show SEM and XRD data after AgNPs loading.
According to the Reviewer’s comment, we supplemented the distribution of Ag on the nanotubes using SEM and EDX mapping techniques.
The distribution of Ag over the nanotubes is shown in Figure S1. Most Ag nanoparticles were distributed on the TNA surface. During the preparation of SEM samples, Ag may aggregate during the drying process. The SERS test can be completed in 1 min; during this time, the Ag nanoparticles are moving, and their distribution is more uniform than that shown in Figure S1.
We also supplemented the discussion on the influences of distribution and load on the SERS effect.
Thess revised sentences appear in the Results and Discussion section on lines 97-101 on page 3 of the revised manuscript (when the "Track Changes" is displayed).
Experimental part: please, add parameters for Raman spectra collection (accumulation time and so on), what is power of UV lamp, which you used for degradation (this is an important parameter).
According to the Reviewer’s comments, we added parameters for Raman spectrum collection (including accumulation time, grating, objective lens strength, etc.) and the power of the UV lamps used for degradation.
All Raman spectra were measured with a 600 gr/mm grating using a 50×, 0.50 NA (OLYMPUS LMPlanFLN, Japan) long-working-distance (LWD) microscope objective. The laser power reaching the sample surface was approximately 5 mW. The acquisition time of Raman spectra was 30 s for each window.
The UV degradation experiment was performed under irradiation with a 254 nm UV lamp with a power of 8 W.
These revised sentences appear in the Materials and Methods section on lines 241-244 and lines 246-248 on page 7 of the revised manuscript (when the "Track Changes" is displayed).
Sensor: please, provide LOD determination according to IUPAC recommendations.
According to the Reviewer’s comments, we provided the LOD determination according to IUPAC recommendations in Figure 5C. We also added detailed explanations according to the LOD results.
The standard curve of the AHMA-FA solution obeys the equation Y = 5.1399 + 0.2701 X, which shows a linear relationship and has a correlation coefficient of 0.9639. The limit of detection (LOD) was determined according to the IUPAC recommendations (the minimally acceptable signal intensity must be three times greater than the standard deviation of the blank signal). As depicted in Figure 5, this intensity approximately corresponds to 10−6.92 M (i.e., 1.21´10−7 M). Finally, in terms of reproducibility and repeatability, SERS measurements were obtained on 20 spots randomly selected on the Ag-TNA substrates (see Figure S2 in the Supplementary Material).
These revised sentences appear in the Abstract on lines 23 and in the Results and Discussion section on lines 155-162 on page 5 of the revised manuscript (when the "Track Changes" is displayed).
Moderate English changes required
According to the Reviewer’s comment, we have revised our manuscript. In addition, the revised manuscript was edited for proper English language, grammar, punctuation, spelling, and overall style by two of the highly qualified native English speaking editors at American Journal Experts (www.aje.com).
The editorial certificate for the revised manuscript is attached as supporting information for the Editor.
These revisions appear in the revised manuscript (when the "Track Changes" is displayed).

Round 2
Reviewer 3 Report
Authors contributes to all comments